# Conformal Prediction Under Covariate Shift

**Ryan J. Tibshirani**
Department of Statistics
Machine Learning Department
Carnegie Mellon University
Pittsburgh PA, 15213
ryantibs@cmu.edu

**Rina Foygel Barber**
Department of Statistics
University of Chicago
Chicago, IL 60637
rina@uchicago.edu

**Emmanuel J. Candès**
Department of Statistics
Department of Mathematics
Stanford University
Stanford CA, 94305
candes@stanford.edu

**Aaditya Ramdas**
Department of Statistics
Machine Learning Department
Carnegie Mellon University
Pittsburgh PA, 15213
aramdas@cmu.edu

## Abstract

We extend conformal prediction methodology beyond the case of exchangeable data. In particular, we show that a weighted version of conformal prediction can be used to compute distribution-free prediction intervals for problems in which the test and training covariate distributions differ, but the likelihood ratio between the two distributions is known—or, in practice, can be estimated accurately from a set of unlabeled data (test covariate points). Our weighted extension of conformal prediction also applies more broadly, to settings in which the data satisfies a certain weighted notion of exchangeability. We discuss other potential applications of our new conformal methodology, including latent variable and missing data problems.

## 1 Introduction

Let $(X_i, Y_i) \in \mathbb{R}^d \times \mathbb{R}$, $i = 1, \ldots, n$ denote training data, assumed to be i.i.d. from an arbitrary distribution $P$. Given a desired coverage rate $1 - \alpha \in (0, 1)$, consider the problem of constructing a band $\widehat{C}_n : \mathbb{R}^d \to \{\text{subsets of } \mathbb{R}\}$, based on the training data such that, for a new i.i.d. point $(X_{n+1}, Y_{n+1})$,

$$\mathbb{P}\left\{ Y_{n+1} \in \widehat{C}_n(X_{n+1}) \right\} \geq 1 - \alpha, \tag{1}$$

where this probability is taken over the $n + 1$ points $(X_i, Y_i)$, $i = 1, \ldots, n + 1$ (the $n$ training points and the test point). Crucially, we will require (1) to hold with no assumptions whatsoever on the underlying distribution $P$.

*Conformal prediction*, a framework pioneered by Vladimir Vovk and colleagues in the 1990s, provides a means for achieving this goal, relying only on exchangeablility of the training and test data. The definitive reference is the book by Vovk et al. [2005]; see also Shafer and Vovk [2008], Vovk et al. [2009], Vovk [2013], Burnaev and Vovk [2014], and http://www.alrw.net for an often-updated list of conformal prediction work by Vovk and colleagues. Moreover, see Lei and Wasserman [2014], Lei et al. [2018] for recent developments in the areas of nonparametric and high-dimensional regression.

In this work, we extend conformal prediction beyond the setting of exchangeable data, allowing for provably valid inference even when the training and test data are not drawn from the same distribution. We begin by reviewing the basics of conformal prediction, in this section. In Section 2, we describe

an extension of conformal prediction to the setting of covariate shift, and give supporting empirical results. In Section 3, we cover the mathematical details behind our conformal extension. We conclude in Section 4 with a short discussion.

## 1.1 Quantile lemma

Before explaining the basic ideas behind conformal inference (i.e., conformal prediction, we will use these two terms interchangeably), we introduce some notation. We denote by $\mathrm{Quantile}(\beta; F)$ the level $\beta$ quantile of a distribution $F$, i.e., for $Z \sim F$,

$$\mathrm{Quantile}(\beta; F) = \inf\big\{z : \mathbb{P}\{Z \le z\} \ge \beta\big\}.$$

In our use of quantiles, we will allow for distributions $F$ on the augmented real line, $\mathbb{R} \cup \{\infty\}$. For values $v_1, \ldots, v_n$, we write $v_{1:n} = \{v_1, \ldots, v_n\}$ to denote their multiset. Note that this is unordered, and allows for multiple instances of the same element; thus in the present case, if $v_i = v_j$ for $i \ne j$, then this value appears twice in $v_{1:n}$. To denote quantiles of the empirical distribution of the values $v_1, \ldots, v_n$, we abbreviate

$$\mathrm{Quantile}(\beta; v_{1:n}) = \mathrm{Quantile}\bigg(\beta; \frac{1}{n}\sum_{i=1}^{n}\delta_{v_i}\bigg),$$

where $\delta_a$ denotes a point mass at $a$ (i.e., the distribution that places all mass at the value $a$). The next result is a simple but key component underlying conformal prediction. Its proof, as with all proofs in this paper, is deferred to the supplement.

**Lemma 1.** *If $V_1, \ldots, V_{n+1}$ are exchangeable random variables, then for any $\beta \in (0, 1)$, we have*

$$\mathbb{P}\Big\{V_{n+1} \le \mathrm{Quantile}\big(\beta; V_{1:n} \cup \{\infty\}\big)\Big\} \ge \beta.$$

*Furthermore, if ties between $V_1, \ldots, V_{n+1}$ occur with probability zero, then the above probability is upper bounded by $\beta + 1/(n+1)$.*

## 1.2 Conformal prediction

We now return to the regression setting.[1] Denote $Z_i = (X_i, Y_i)$, $i = 1, \ldots, n$. In what follows, we describe the construction of a prediction band satisfying (1), using conformal inference, due to Vovk et al. [2005]. We first choose a score function $\mathcal{S}$, whose arguments consist of a point $(x, y)$, and a multiset $Z$.[2] Informally, a low value of $\mathcal{S}((x, y), Z)$ indicates that the point $(x, y)$ "conforms" to $Z$, whereas a high value indicates that $(x, y)$ is atypical relative to the points in $Z$. For example, we might choose to define $\mathcal{S}$ by

$$\mathcal{S}\big((x, y), Z\big) = |y - \widehat{\mu}(x)|, \tag{2}$$

where $\widehat{\mu} : \mathbb{R}^d \to \mathbb{R}$ is a regression function, fitted by running an algorithm $\mathcal{A}$ on $Z$. Next, at a given $x \in \mathbb{R}^d$, we define $\widehat{C}_n(x)$, the conformal prediction interval[3], by repeating the following procedure for each $y \in \mathbb{R}$. We calculate the *nonconformity scores*

$$V_i^{(x,y)} = \mathcal{S}\big(Z_i, Z_{1:n} \cup \{(x, y)\}\big), \ i = 1, \ldots, n, \quad \text{and} \quad V_{n+1}^{(x,y)} = \mathcal{S}\big((x, y), Z_{1:n} \cup \{(x, y)\}\big), \tag{3}$$

and include $y$ in our prediction interval $\widehat{C}_n(x)$ if

$$V_{n+1}^{(x,y)} \le \mathrm{Quantile}\big(1 - \alpha; V_{1:n}^{(x,y)} \cup \{\infty\}\big),$$

where $V_{1:n}^{(x,y)} = \{V_1^{(x,y)}, \ldots, V_n^{(x,y)}\}$. Importantly, the symmetry in the construction of the nonconformity scores (3) guarantees exact coverage in finite samples. The next theorem summarizes this coverage result. The lower bound is a standard result from the conformal literature, see Vovk et al. [2005]; the upper bound, as far as we know, was first pointed out by Lei et al. [2018].

**Theorem 1** (Vovk et al. 2005, Lei et al. 2018). *Assume that $(X_i, Y_i) \in \mathbb{R}^d \times \mathbb{R}$, $i = 1, \ldots, n+1$ are exchangeable. For any score function $\mathcal{S}$, and any $\alpha \in (0, 1)$, define the conformal band (based on the first $n$ samples) at $x \in \mathbb{R}^d$ by*

$$\widehat{C}_n(x) = \left\{ y \in \mathbb{R} : V_{n+1}^{(x,y)} \leq \mathrm{Quantile}\big(1 - \alpha; V_{1:n}^{(x,y)} \cup \{\infty\}\big) \right\}, \tag{4}$$

*where $V_i^{(x,y)}$, $i = 1, \ldots, n+1$ are as defined in (3). Then $\widehat{C}_n$ satisfies*

$$\mathbb{P}\left\{ Y_{n+1} \in \widehat{C}_n(X_{n+1}) \right\} \geq 1 - \alpha.$$

*Furthermore, if ties between $V_1^{(X_{n+1}, Y_{n+1})}, \ldots, V_{n+1}^{(X_{n+1}, Y_{n+1})}$ occur with probability zero, then this probability is upper bounded by $1 - \alpha + 1/(n+1)$.*

**Remark 1.** Theorem 1 is stated assuming exchangeable samples $(X_i, Y_i)$, $i = 1, \ldots, n+1$, which is weaker than assuming i.i.d. samples. As we will see in what follows, it is possible to relax the exchangeability assumption, under an appropriate modification to the conformal procedure.

**Remark 2.** If we use an appropriate random tie-breaking rule (to determine the rank of $V_{n+1}$ among $V_1, \ldots, V_{n+1}$), then the upper bounds in Lemma 1 and Theorem 1 hold in general (without assuming there are no ties almost surely).

The result in Theorem 1, albeit simple to prove, is quite remarkable. It gives a recipe for distribution-free prediction intervals, having nearly exact coverage, starting from an arbitrary score function $\mathcal{S}$; e.g., absolute residuals defined using a fitted regression function from any base algorithm $\mathcal{A}$, as in (2). For more discussion of conformal prediction, its properties, and its variants, see Vovk et al. [2005], Lei et al. [2018] and references therein.

## 2 Covariate shift

In this paper, we are concerned with settings in which the data $(X_i, Y_i)$, $i = 1, \ldots, n+1$ are no longer exchangeable. Our primary focus will be a setting in which we observe data according to

$$
\begin{aligned}
(X_i, Y_i) &\overset{\text{i.i.d.}}{\sim} P = P_X \times P_{Y|X}, \ i = 1, \ldots, n, \\
(X_{n+1}, Y_{n+1}) &\sim \widetilde{P} = \widetilde{P}_X \times P_{Y|X}, \ \text{independently.}
\end{aligned}
\tag{5}
$$

Notice that the conditional distribution of $Y|X$ is assumed to be the same for both the training and test data. Such a setting is often called *covariate shift* (e.g., see Shimodaira 2000, Quinonero-Candela et al. 2009; see also Remark 4 below for more discussion of this literature). The key realization is the following: if we know the ratio of test to training covariate likelihoods, $d\widetilde{P}_X/dP_X$, then we can still perform a modified of version conformal inference, using a quantile of a suitably weighted empirical distribution of nonconformity scores. The next subsection gives details; following this, we give an empirical demonstration.

### 2.1 Weighted conformal prediction

In conformal prediction, we form a prediction interval by comparing the value of a nonconformity score at a test point to the empirical distribution of nonconformity scores at the training points. In the covariate shift case, where the covariate distributions $P_X, \widetilde{P}_X$ in our training and test sets differ, we will now weight each nonconformity score $V_i^{(x,y)}$ (measuring how well $Z_i = (X_j, Y_j)$ conforms to the other points) by a probability proportional to the likelihood ratio $w(X_i) = d\widetilde{P}_X(X_i)/dP_X(X_i)$. Therefore, we will no longer be interested in the empirical distribution $\frac{1}{n+1} \sum_{i=1}^{n} \delta_{V_i^{(x,y)}} + \frac{1}{n+1} \delta_\infty$, as in Theorem 1, but rather, a weighted version

$$\sum_{i=1}^{n} p_i^w(x) \delta_{V_i^{(x,y)}} + p_{n+1}^w(x) \delta_\infty,$$

where the weights are defined by

$$p_i^w(x) = \frac{w(X_i)}{\sum_{j=1}^{n} w(X_j) + w(x)}, \ i = 1, \ldots, n, \quad \text{and} \quad p_{n+1}^w(x) = \frac{w(x)}{\sum_{j=1}^{n} w(X_j) + w(x)}. \tag{6}$$

Due this careful weighting, draws from the discrete distribution in the second to last display resemble nonconformity scores computed on the test population, and thus, they "look exchangeable" with the nonconformity score at our test point. Our main result below formalizes these claims.

**Corollary 1.** *Assume data from the model* (5). *Assume* $\widetilde{P}_X$ *is absolutely continuous with respect to* $P_X$, *and denote* $w = \mathrm{d}\widetilde{P}_X/\mathrm{d}P_X$. *For any score function* $\mathcal{S}$, *and any* $\alpha \in (0,1)$, *define for* $x \in \mathbb{R}^d$,

$$\widehat{C}_n(x) = \left\{ y \in \mathbb{R} : V_{n+1}^{(x,y)} \le \mathrm{Quantile}\left(1-\alpha; \sum_{i=1}^n p_i^w(x)\delta_{V_i^{(x,y)}} + p_{n+1}^w(x)\delta_\infty\right)\right\}, \quad (7)$$

*where* $V_i^{(x,y)}$, $i = 1, \ldots, n+1$ *are as defined in* (3), *and* $p_i^w$, $i = 1, \ldots, n+1$ *are as defined in* (6). *Then* $\widehat{C}_n$ *satisfies*

$$\mathbb{P}\left\{Y_{n+1} \in \widehat{C}_n(X_{n+1})\right\} \ge 1 - \alpha.$$

Corollary 1 is a special case of a more general result presented later in Theorem 2, which extends conformal inference to a setting in which the data are what we call *weighted exchangeable*.

**Remark 3.** The same result as in Corollary 1 holds if $w \propto \mathrm{d}\widetilde{P}_X/\mathrm{d}P_X$, i.e., with unknown normalization constant, because this constant cancels out in the calculation of probabilities in (6).

**Remark 4.** Though the basic premise of covariate shift—and certainly the techniques employed in addressing it—are related to much older ideas in statistics, the specific setup in (5) has recently generated great interest in machine learning: e.g., see Sugiyama and Muller [2005], Sugiyama et al. [2007], Quinonero-Candela et al. [2009], Agarwal et al. [2011], Wen et al. [2014], Reddi et al. [2015], Chen et al. [2016] and references therein). The focus is usually on correcting estimators, model evaluation, or model selection approaches to account for covariate shift. Correcting distribution-free prediction intervals, as we examine in this work, is (as far as we know) a new contribution. As one might expect, the likelihood ratio $\mathrm{d}\widetilde{P}_X/\mathrm{d}P_X$, a key component of our conformal construction in Corollary 1, also plays a critical role in much of the literature on covariate shift.

## 2.2 Airfoil data example

We demonstrate conformal prediction in the covariate shift setting using an empirical example. We consider the airfoil data set from the UCI Machine Learning Repository [Dua and Graff, 2019], which has $N = 1503$ observations of a response $Y$ (scaled sound pressure level of NASA airfoils), and a covariate $X$ with $d = 5$ dimensions (log frequency, angle of attack, chord length, free-stream velocity, and suction side log displacement thickness). For efficiency, we use a variant of conformal prediction called *split conformal prediction* [Papadopoulos et al., 2002, Lei et al., 2015], which we extend to the covariate shift case in the same way (using weighted quantiles); see the supplement. For R code to reproduce the results that follow, see `http://www.github.com/ryantibs/conformal/`.

**Creating training data, test data, and covariate shift.** We repeated an experiment for 5000 trials, where for each trial we randomly partitioned the data $\{(X_i, Y_i)\}_{i=1}^N$ into two sets $D_{\text{train}}, D_{\text{test}}$, and also constructed a covariate shift test set $D_{\text{shift}}$, which have the following roles.

- $D_{\text{train}}$, containing 50% of the data, is our training set, i.e., $(X_i, Y_i)$, $i = 1, \ldots, n$, used to compute conformal prediction intervals (using the split conformal variant).

- $D_{\text{test}}$, containing 50% of the data, is our test set (as these data points are exchangeable with those in $D_{\text{train}}$, there is no covariate shift in this test set).

- $D_{\text{shift}}$ is a second test set, constructed to simulate covariate shift, by sampling 25% of the points from $D_{\text{test}}$ with replacement, with probabilities proportional to

$$w(x) = \exp(x^T\beta), \quad \text{where} \quad \beta = (-1, 0, 0, 0, 1). \quad (8)$$

As the original data points $D_{\text{train}} \cup D_{\text{test}}$ can be seen as draws from the same underlying distribution, we can view $w(x)$ as the likelihood ratio of covariate distributions between the test set $D_{\text{shift}}$ and training set $D_{\text{train}}$. Note that the test covariate distribution $\widetilde{P}_X$, which satisfies $\mathrm{d}\widetilde{P}_X \propto \exp(x^T\beta)\mathrm{d}P_X$ as we have defined it here, is called an *exponential tilting* of the training covariate distribution $P_X$. The supplement displays kernel density estimates fit to the airfoil data set, pre and post exponential tilting, to visualize the differences in the covariate distributions.

**Loss of coverage of ordinary conformal prediction under covariate shift.** First, we examine the performance of ordinary split conformal prediction. The nominal coverage level was set to be 90% (meaning $\alpha = 0.1$), here and throughout. The results are displayed in the top row of Figure 1.

In each of the 5000 trials, we computed the empirical coverage from the split conformal intervals over points in the test sets, and the histograms show the distribution of these empirical coverages over the trials. We see that for the original test data $D_{\text{test}}$ (no covariate shift, shown in red), split conformal works as expected, with the average of the empirical coverages (over the 5000 trials) being 90.2%; for the nonuniformly subsampled test data $D_{\text{shift}}$ (covariate shift, in blue), split conformal considerably undercovers, with its average coverage being 82.2%.

**Coverage of weighted conformal prediction with oracle weights.** Next, displayed in the middle row of Figure 1, we consider weighted split conformal prediction, to cover the points in $D_{\text{shift}}$ (shown in orange). At the moment, we assume oracle knowledge of the true weight function $w$ in (8) needed to calculate the probabilities in (6). We see that this brings the coverage back to the desired level, with the average coverage being 90.8%. However, the histogram is more dispersed than it is when there is no covariate shift (compare to the top row, in red). This is because, by using a quantile of the weighted empirical distribution of nonconformity scores, we are relying on a reduced "effective sample size". Given training points $X_1, \ldots, X_n$, and a likelihood ratio $w$ of test to training covariate distributions, a popular heuristic formula from the covariate shift literature for the effective sample size of $X_1, \ldots, X_n$ is [Gretton et al., 2009, Reddi et al., 2015]:

$$\widehat{n} = \frac{[\sum_{i=1}^{n} |w(X_i)|]^2}{\sum_{i=1}^{n} |w(X_i)|^2} = \frac{\|w(X_{1:n})\|_1^2}{\|w(X_{1:n})\|_2^2},$$

where we abbreviate $w(X_{1:n}) = (w(X_1), \ldots, w(X_n)) \in \mathbb{R}^n$. To compare weighted conformal prediction against the unweighted method at the same effective sample size, in each trial, we ran unweighted split conformal on the original test set $D_{\text{test}}$, but we used only $\widehat{n}$ subsampled points from $D_{\text{train}}$ to compute the quantile of nonconformity scores. The results (the middle row of Figure 1, in purple) line up closely with those from weighted conformal, which shows that the overdispersion in the coverage histogram from the latter is fully explained by the reduced effective sample size.

**Coverage of weighted conformal with estimated weights.** Denote by $X_1, \ldots, X_n$ the covariate points in $D_{\text{train}}$ and by $X_{n+1}, \ldots, X_{n+m}$ the covariate points in $D_{\text{shift}}$. Here we describe how to estimate $w = \mathrm{d}\widetilde{P}_X/\mathrm{d}P_X$, the likelihood ratio of interest, by applying logistic regression or random forests (more generally, any classifier that outputs estimated probabilities of class membership) to the feature-class pairs $(X_i, C_i)$, $i = 1, \ldots, n + m$, where $C_i = 0$ for $i = 1, \ldots, n$ and $C_i = 1$ for $i = n + 1, \ldots, n + m$. Noting that

$$\frac{\mathbb{P}(C = 1|X = x)}{\mathbb{P}(C = 0|X = x)} = \frac{\mathbb{P}(C = 1)}{\mathbb{P}(C = 0)} \frac{\mathrm{d}\widetilde{P}_X}{\mathrm{d}P_X}(x),$$

we can take the conditional odds ratio $w(x) = \mathbb{P}(C = 1|X = x)/\mathbb{P}(C = 0|X = x)$ as an equivalent representation for the oracle weight function (since we only need to know the likelihood ratio up to a proportionality constant, recall Remark 3). Therefore, if $\widehat{p}(x)$ is an estimate of $\mathbb{P}(C = 1|X = x)$ obtained by fitting a classifier to the data $(X_i, C_i)$, $i = 1, \ldots, n + m$, then we can use

$$\widehat{w}(x) = \frac{\widehat{p}(x)}{1 - \widehat{p}(x)} \tag{9}$$

as our estimated weight function for the calculation of probabilities (6) that are needed for conformal prediction. There is in fact a sizeable literature on density ratio estimation, and the method just describe falls into a class called *probabilistic classification* approaches; two other classes are based on moment matching, and minimization of $\phi$-divergences (e.g., Kullback-Leibler divergence). For a comprehensive review of these approaches, and supporting theory, see Sugiyama et al. [2012].

The bottom row of Figure 1 shows the results from using weighted split conformal prediction to cover the points in $D_{\text{shift}}$, where the weight function $\widehat{w}$ has been estimated as in (9), using logistic regression (in gray) and random forests[4] (in green) to fit the class probability function $\widehat{p}$. Note that logistic regression is well-specified in this example, as it assumes the log odds is a linear function of $x$, which is exactly as in (8). Random forests, of course, allows more flexibility in the fitted model.

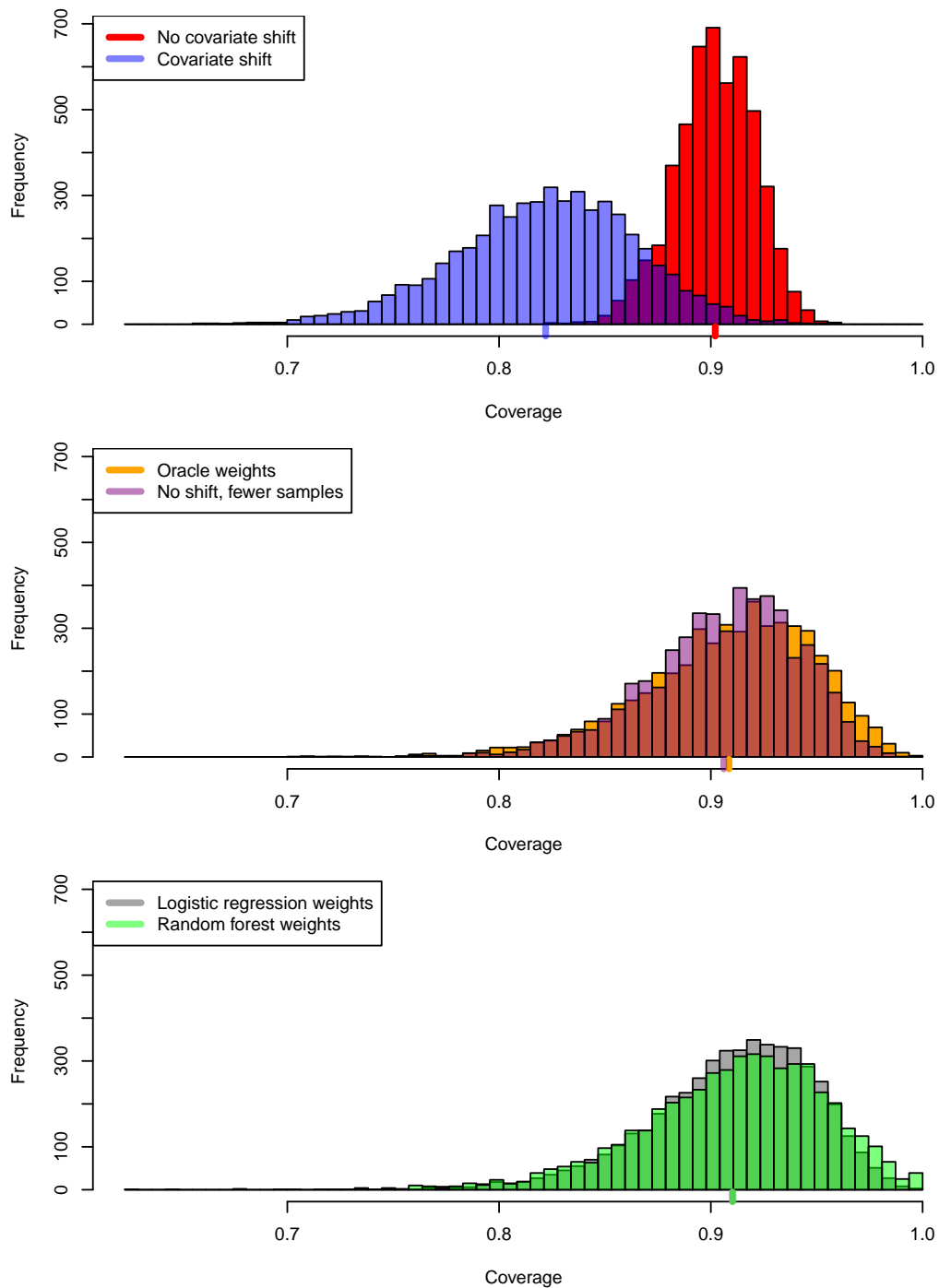

Figure 1: Empirical coverages of conformal prediction intervals, computed using 5000 different random splits of the airfoil data set. The averages of empirical coverages in each histogram are marked on the x-axis.

Both classification approaches deliver weights that translate into good average coverage, being 91.0% for each approach. Furthermore, their histograms are only a little more dispersed than that for the oracle weights (middle row, in orange). For more simulation results, see the supplement.

# 3 Weighted exchangeability

In this section, we develop a general result on conformal prediction for settings in which the data satisfy what we call *weighted exchangeability*. First we precisely define this concept, then we extend Lemma 1 to this new setting, and extend conformal prediction as well.

## 3.1 Generalizing exchangeability

We first define a generalized notion of exchangeability.

**Definition 1.** We call random variables $V_1, \ldots, V_n$ *weighted exchangeable*, with weight functions $w_1, \ldots, w_n$, if the density[5] $f$ of their joint distribution can be factorized as

$$f(v_1, \ldots, v_n) = \prod_{i=1}^{n} w_i(v_i) \cdot g(v_1, \ldots, v_n),$$

where $g$ does not depend on the ordering of its inputs, i.e., $g(v_{\sigma(1)}, \ldots, v_{\sigma(n)}) = g(v_1, \ldots, v_n)$ for any permutation $\sigma$ of $1, \ldots, n$.

Clearly, weighted exchangeability with weight functions $w_i \equiv 1$, $i = 1, \ldots, n$ reduces to ordinary exchangeability. Furthermore, independent draws (where all marginal distributions are absolutely continuous with respect to, say, the first one) are always weighted exchangeable, with weight functions given by the appropriate Radon-Nikodym derivatives, i.e., likelihood ratios. This is stated next; the proof follows directly from Definition 1 and is omitted.

**Lemma 2.** *Let $Z_i \sim P_i$, $i = 1, \ldots, n$ be independent draws, where each $P_i$ is absolutely continuous with respect to $P_1$, for $i \geq 2$. Then $Z_1, \ldots, Z_n$ are weighted exchangeable, with weight functions $w_1 \equiv 1$, and $w_i = \mathrm{d}P_i/\mathrm{d}P_1$, $i \geq 2$.*

Lemma 2 highlights an important special case (which we note, includes the covariate shift model in (5)). But it is worth being clear that our definition of weighted exchangeability encompasses more than independent sampling, and allows for a nontrivial dependency structure between the variables.

## 3.2 Generalizing conformal prediction

Now we give a weighted generalization of Lemma 1.

**Lemma 3.** *Let $Z_i$, $i = 1, \ldots, n+1$ be weighted exchangeable, with weight functions $w_1, \ldots, w_{n+1}$. Let $V_i = \mathcal{S}(Z_i, Z_{1:(n+1)})$, for $i = 1, \ldots, n+1$, and $\mathcal{S}$ is an arbitrary score function. Define*

$$p_i^w(z_1, \ldots, z_{n+1}) = \frac{\sum_{\sigma:\sigma(n+1)=i} \prod_{j=1}^{n+1} w_j(z_{\sigma(j)})}{\sum_{\sigma} \prod_{j=1}^{n+1} w_j(z_{\sigma(j)})}, \ i = 1, \ldots, n+1, \tag{10}$$

*where the summations are taken over permutations $\sigma$ of the numbers $1, \ldots, n+1$. Then for any $\beta \in (0, 1)$,*

$$\mathbb{P}\left\{ V_{n+1} \leq \mathrm{Quantile}\left(\beta; \sum_{i=1}^{n} p_i^w(Z_1, \ldots, Z_{n+1})\delta_{V_i} + p_{n+1}^w(Z_1, \ldots, Z_{n+1})\delta_{\infty}\right)\right\} \geq \beta.$$

**Remark 5.** When $V_1, \ldots, V_{n+1}$ are exchangeable, we have $w_i \equiv 1$ for $i = 1, \ldots, n$, and so $p_i^w \equiv 1$ for $i = 1, \ldots, n$. Note that, in this special case, the lower bound in Lemma 3 reduces to the ordinary unweighted lower bound in Lemma 1. On the other hand, obtaining a meaningful upper bound on the probability in question in Lemma 3, as was done in Lemma 1 (when we assume almost surely no ties), does not seem possible without further conditions on the weight functions. This is because the largest jump in the cumulative distribution function of $V_{n+1}|E_z$ is of size $\max_{i=1,\ldots,n+1} p_i^w(z_1, \ldots, z_{n+1})$, which can potentially be very large; in the unweighted case, this jump is always of size $1/(n+1)$.

A weighted version of conformal prediction follows immediately from Lemma 3.

**Theorem 2.** *Assume that $Z_i = (X_i, Y_i) \in \mathbb{R}^d \times \mathbb{R}$, $i = 1, \ldots, n+1$ are weighted exchangeable with weight functions $w_1, \ldots, w_{n+1}$. For any score function $\mathcal{S}$, and any $\alpha \in (0,1)$, define the weighted conformal band (based on the first $n$ samples) at a point $x \in \mathbb{R}^d$ by*

$$\widehat{C}_n(x) = \left\{ y \in \mathbb{R} : V_{n+1}^{(x,y)} \leq \mathrm{Quantile}\left( 1 - \alpha; \sum_{i=1}^{n} p_i^w\big(Z_1, \ldots, Z_n, (x,y)\big) \delta_{V_i^{(x,y)}} + \right.\right.$$

$$\left.\left. p_{n+1}^w\big(Z_1, \ldots, Z_n, (x,y)\big) \delta_\infty \right) \right\}, \quad (11)$$

*where $V_i^{(x,y)}$, $i = 1, \ldots, n+1$ are as defined in (3), and $p_i^w$, $i = 1, \ldots, n+1$ are as defined in (10). Then $\widehat{C}_n$ satisfies*

$$\mathbb{P}\left\{ Y_{n+1} \in \widehat{C}_n(X_{n+1}) \right\} \geq 1 - \alpha.$$

Observe that Corollary 1 follows by taking $w_i \equiv 1$ for $i = 1, \ldots, n$, and $w_{n+1}((x,y)) = w(x)$.

## 4 Discussion

We described an extension of conformal prediction to handle weighted exchangeable data, covering exchangeable data, and independent (but not identically distributed) data, as special cases. In general, the new weighted methodology requires computing quantiles of a weighted discrete distribution of nonconformity scores, which is combinatorially hard. But the computations simplify dramatically for a case of significant practical interest, where the test covariate distribution $\widetilde{P}_X$ differs from the training covariate distribution $P_X$ by a known likelihood ratio $d\widetilde{P}_X/dP_X$ (and the conditional distribution $P_{Y|X}$ remains unchanged). In this case, called covariate shift, the new weighted conformal prediction methodology is just as easy, computationally, as ordinary conformal prediction. When the likelihood ratio $d\widetilde{P}_X/dP_X$ is not known, it can be estimated given access to unlabeled data (test covariate points), which we showed empirically, on a low-dimensional example, can still yield correct coverage.

Beyond the setting of covariate shift that we have focused on (as the main application in this paper), our weighted conformal methodology can be applied to several other closely related settings, where ordinary conformal prediction will not directly yield correct coverage. We discuss two such settings below; a third, on approximate conditional inference, is discussed the supplement.

**Graphical models with covariate shift.** Assume that the training data $(Z, X, Y) \sim P$ obeys the Markovian structure $Z \to X \to Y$. As an example, to make matters concrete, suppose that $Z$ is a low-dimensional covariate (such as ancenstry information), $X$ is a high-dimensional set of features for a person (such as genetic measurements), and $Y$ is a real-valued outcome of interest (such as life expectancy). Suppose that on the test data $(Z, X, Y) \sim \widetilde{P}$, the distribution of $Z$ has changed, causing a change in the distribution of $X$, and thus causing a change in the distribution of the unobserved $Y$ (however the distribution of $X|Z$ is unchanged). One plausible solution to this problem would be to just ignore $Z$ in both training and test sets, and run weighted conformal prediction on only $(X, Y)$, treating this like a usual covariate shift problem. But, as $X$ is high-dimensional, this would require estimating a ratio of two high-dimensional densities, which would be difficult. Since $Z$ is low-dimensional, we can instead estimate the weights by estimating the likelihood ratio of $Z$ between test and training sets, which follows because for the joint covariate $(Z, X)$,

$$\frac{\widetilde{P}_{Z,X}(z,x)}{P_{Z,X}(z,x)} = \frac{\widetilde{P}_Z(z) P_{X|Z=z}(x)}{P_Z(z) P_{X|Z=z}(x)} = \frac{\widetilde{P}_Z(z)}{P_Z(z)}.$$

This may be a more tractable quantity to estimate for the purpose of weighted conformal inference. These ideas may be generalized to more complex graphical settings, which we leave to future work.

**Missing covariates with known summaries.** As another concrete example, suppose that hospital A has collected a private training data set $(Z, X, Y) \sim P^A$ where $Z \in \{0, 1\}$ is a sensitive patient covariate, $X$ represents other covariates, and $Y$ is a real-valued response that is expensive to measure. Suppose that hospital B also has its own data set, but in order to save money and not measure the responses for their patients, it asks hospital A for help to produce prediction intervals for these

responses. Instead of sharing the collected data $(Z, X) \sim P^B$ for each patient with hospital A, due to privacy concerns, hospital B only provides hospital A with the $X$ covariate for each patient, along with a summary statistic for $Z$, representing the fraction of $Z$ values that equal one (more accurately, the probability of drawing a patient with $Z = 1$ from their underlying patient population). Assume that $P_{X|Z}^A = P_{X|Z}^B$ (e.g., if $Z$ is the sex of the patient, then this assumes there is one joint distribution on $X$ for males and one for females, which does not depend on the hospital). The likelihood ratio of covariate distributions thus again reduces to calculating the likelihood ratio of $Z$ between $P^B$ and $P^A$, which we can easily do, and use weighted conformal prediction.

**Towards local conditional coverage?**     We finish by desciribing how our weighted conformal methodology can be used to construct prediction bands with certain a approximate notion of conditional coverage. Given i.i.d. $(X_i, Y_i)$, $i = 1, \ldots, n+1$, consider, instead of the original goal (1),

$$\mathbb{P}\Big\{Y_{n+1} \in \widehat{C}_n(x_0) \,\Big|\, X_{n+1} = x_0\Big\} \geq 1 - \alpha. \tag{12}$$

This is (exact) conditional coverage at $x_0 \in \mathbb{R}^d$. As it turns out, asking for (12) to hold at $P_X$-almost every $x_0 \in \mathbb{R}^d$, and for all distributions $P$ is far too strong: Vovk [2012], Lei and Wasserman [2014] prove that any method with such a property must yield an interval $\widehat{C}_n(x_0)$ with infinite expected length at any non-atom point $x_0$, for any underlying distribution $P$. Thus we must relax (12) and seek some notion of approximate conditional coverage, if we hope to achieve it with a nontrivial prediction band. Some relaxations were recently considered in Barber et al. [2019], most of which were also impossible to achieve in a nontrivial way. A different, natural relaxation of (12) is

$$\frac{\int \mathbb{P}\big(Y_{n+1} \in \widehat{C}_n(x_0) \,|\, X_{n+1} = x\big) K\big(\frac{x-x_0}{h}\big) \, \mathsf{d}P_X(x)}{\int K\big(\frac{x-x_0}{h}\big) \, \mathsf{d}P_X(x)} \geq 1 - \alpha, \tag{13}$$

where $K$ is kernel function and $h > 0$ is bandwidth parameter. Here we are asking for a prediction band whose average conditional coverage, in some locally-weighted sense around $x$, is at least $1 - \alpha$. We can equivalently write (13) as

$$\mathbb{P}\Big\{Y_{n+1} \in \widehat{C}_n(x_0) \,\Big|\, X_{n+1} = x_0 + h\omega\Big\} \geq 1 - \alpha, \tag{14}$$

where the probability is taken over the $n+1$ data points and an independent draw $\omega$ from a distribution whose density is proportional to $K$. As we can see from (13) (or (14)), this kind of locally-weighted guarantee should be close to a guarantee on conditional coverage, when the bandwidth $h$ is small.

In order to achieve (13) in a distribution-free manner, we can invoke the weighted conformal inference methodology. In particular, note that we can once more rewrite (14) as

$$\mathbb{P}_{x_0}\Big\{Y_{n+1} \in \widehat{C}_n(\widetilde{X}_{n+1})\Big\} \geq 1 - \alpha, \tag{15}$$

where $\mathbb{P}_{x_0}$ integrates over training points $(X_i, Y_i)$, $i = 1, \ldots, n$ i.i.d. from $P = P_X \times P_{Y|X}$ and an independent test point $(\widetilde{X}_{n+1}, Y_{n+1})$, from $\widetilde{P} = \widetilde{P}_X \times P_{Y|X}$, where $\mathsf{d}\widetilde{P}_X/\mathsf{d}P_X \propto K((\cdot - x_0)/h)$. Note that this precisely fits into the covariate shift setting (5). To be explicit, for any score function $\mathcal{S}$, and any $\alpha \in (0,1)$, given a center point $x_0 \in \mathbb{R}^d$ of interest, define

$$\widehat{C}_n(x) = \left\{y \in \mathbb{R} : V_{n+1}^{(x,y)} \leq \text{Quantile}\left(1 - \alpha; \frac{\sum_{i=1}^n K\big(\frac{X_i - x_0}{h}\big)\delta_{V_i^{(x,y)}} + K\big(\frac{x-x_0}{h}\big)\delta_\infty}{\sum_{i=1}^n K\big(\frac{X_i - x_0}{h}\big) + K\big(\frac{x-x_0}{h}\big)}\right)\right\},$$

where $V_i^{(x,y)}$, $i = 1, \ldots, n+1$, are as in (3). Then by Corollary 1,

$$\mathbb{P}_{x_0}\Big\{Y_{n+1} \in \widehat{C}_n(X_{n+1}; x_0)\Big\} \geq 1 - \alpha. \tag{16}$$

This is "almost" of the desired form (15) (equivalently (13), or (14)), except for one critical caveat. The band $\widehat{C}_n(\,\cdot\,; x_0)$ in (16) was constructed based on knowing the center point $x_0$ *in advance*. If we were to ask for local conditional coverage at a new point $x_0$, then the entire band $\widehat{C}_n(\,\cdot\,; x_0)$ must change (must be recomputed) in order to accommodate the new guarantee.

**Acknowledgements.**     The authors thank the American Institute of Mathematics for supporting and hosting our collaboration. R.F.B. was partially supported by the National Science Foundation under grant DMS-1654076 and by an Alfred P. Sloan fellowship. E.J.C. was partially supported by the Office of Naval Research under grant N00014-16-1-2712, by the National Science Foundation under grant DMS-1712800, and by a generous gift from TwoSigma. R.J.T. was partially supported by the National Science Foundation under grant DMS-1554123.

## Footnotes

[1]Throughout this paper, we focus on regression, where the response $Y$ is continuous, for simplicity. The same ideas can be applied to classification, where $Y$ is discrete.

[2]We emphasize that by defining $Z$ to be a multiset, we are treating its points as unordered. Hence, to be perfectly explicit, the score function $\mathcal{S}$ cannot accept the points in $Z$ in any particular order, and it must take them in as unordered. The same is true of the base algorithm $\mathcal{A}$ used to define the fitted regression function $\widehat{\mu}$, in the choice of absolute residual score function (2).

[3]For convenience, throughout, we will refer to $\widehat{C}_n(x)$ as an "interval", even though this may actually be a union of multiple nonoverlapping intervals. Similarly, for simplicity, we will refer to $\widehat{C}_n$ as a "band".

[4] In the random forests approach, we clipped the estimated test class probability $\widehat{p}(x)$ to lie in between 0.01 and 0.99, to prevent the estimated weight (likelihood ratio) $\widehat{w}(x)$ from being infinite. Without clipping, the estimated probability of being in the test class was sometimes exactly 1 (this occurred in about 2% of the cases encountered over all 5000 repetitions), resulting in an infinite weight, and causing numerical issues.

[5]More generally, $f$ may be the Radon-Nikodym derivative with respect to an arbitrary base measure.

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
