[Supplementary Material]

# Supplement to "Conformal Prediction Under Covariate Shift"

**Ryan J. Tibshirani**
Department of Statistics
Machine Learning Department
Carnegie Mellon University
Pittsburgh PA, 15213
ryantibs@cmu.edu

**Rina Foygel Barber**
Department of Statistics
University of Chicago
Chicago, IL 60637
rina@uchicago.edu

**Emmanuel J. Candès**
Department of Statistics
Department of Mathematics
Stanford University
Stanford CA, 94305
candes@stanford.edu

**Aaditya Ramdas**
Department of Statistics
Machine Learning Department
Carnegie Mellon University
Pittsburgh PA, 15213
aramdas@cmu.edu

This document give additional details, proofs, and simulation results for "Conformal Prediction Under Covariate Shift".

## A.1    Proof of Lemma 1

Consider the following useful fact about quantiles of a discrete distribution $F$, having support points $a_1, \ldots, a_k \in \mathbb{R}$: denoting $q = \text{Quantile}(\beta; F)$, if we reassign the points $a_i > q$ to arbitrary values that are strictly larger than $q$, yielding a new distribution $\widetilde{F}$, then the level $\beta$ quantile is unchanged, $\text{Quantile}(\beta; F) = \text{Quantile}(\beta; \widetilde{F})$. Using this fact,

$$V_{n+1} > \text{Quantile}\big(\beta; V_{1:n} \cup \{\infty\}\big) \iff V_{n+1} > \text{Quantile}\big(\beta; V_{1:(n+1)}\big),$$

or equivalently,

$$V_{n+1} \leq \text{Quantile}\big(\beta; V_{1:n} \cup \{\infty\}\big) \iff V_{n+1} \leq \text{Quantile}\big(\beta; V_{1:(n+1)}\big). \qquad (A.1)$$

Moreover, it is straightforward to check that

$$V_{n+1} \leq \text{Quantile}\big(\beta; V_{1:(n+1)}\big) \iff V_{n+1} \text{ is among the } \lceil \beta(n+1) \rceil \text{ smallest of } V_1, \ldots, V_{n+1}.$$

By exchangeability, the latter event occurs with probability at least $\lceil \beta(n+1) \rceil / (n+1) \geq \beta$, which establishes the lower bound; when there are almost surely no ties, it holds with probability exactly $\lceil \beta(n+1) \rceil / (n+1) \leq \beta + 1/(n+1)$, which proves the upper bound.

## A.2    Proof of Theorem 1

To lighten notation, abbreviate $V_i = V_i^{(X_{n+1}, Y_{n+1})}$, $i = 1, \ldots, n+1$. Observe

$$Y_{n+1} \in \widehat{C}_n(X_{n+1}) \iff V_{n+1} \leq \text{Quantile}\big(1 - \alpha; V_{1:n} \cup \{\infty\}\big).$$

By the symmetric construction of the nonconformity scores in (3),

$$(Z_1, \ldots, Z_{n+1}) \overset{d}{=} (Z_{\sigma(1)}, \ldots, Z_{\sigma(n+1)}) \iff (V_1, \ldots, V_{n+1}) \overset{d}{=} (V_{\sigma(1)}, \ldots, V_{\sigma(n+1)}),$$

for any permutation $\sigma$ of the numbers $1, \ldots, n+1$. Therefore, as $Z_1, \ldots, Z_{n+1}$ are exchangeable, so are $V_1, \ldots, V_{n+1}$, and applying Lemma 1 gives the result.

## A.3 Split conformal prediction

In general, constructing a conformal prediction band can be computationally intensive, though this of course depends on the choice of score function. Consider the use of absolute residuals as in (2). To compute the nonconformity scores in (3), we must first run our base algorithm $\mathcal{A}$ on the data set $Z_{1:n} \cup \{(x,y)\}$ to produce a fitted regression function $\widehat{\mu}$, and then calculate

$$V_i^{(x,y)} = |Y_i - \widehat{\mu}(X_i)|, \ i = 1, \ldots, n, \quad \text{and} \quad V_{n+1}^{(x,y)} = |y - \widehat{\mu}(x)|.$$

As the formation of the conformal set in (4) (ordinary case) or (7) (covariate shift case) requires us to do this for each $x \in \mathbb{R}^d$ and $y \in \mathbb{R}$ (which requires refitting $\widehat{\mu}$ each time), this can clearly become computationally burdernsome.

A fast alternative, known as *split conformal prediction* [Papadopoulos et al., 2002, Lei et al., 2015], resolves this issue by taking the score function $\mathcal{S}$ to be defined using absolute residuals with respect to a *fixed* regression function, typically, one that has been trained on an preliminary data set. Denote by $(X_1^0, Y_1^0), \ldots, (X_{n_0}^0, Y_{n_0}^0)$ this preliminary data set, used for fitting the regression function $\mu_0$, and consider the score function

$$\mathcal{S}\big((x,y), Z\big) = |y - \mu_0(x)|.$$

Given data $(X_1, Y_1), \ldots, (X_n, Y_n)$, independent of $(X_1^0, Y_1^0), \ldots, (X_{n_0}^0, Y_{n_0}^0)$, we calculate

$$V_i^{(x,y)} = |Y_i - \mu_0(X_i)|, \ i = 1, \ldots, n, \quad \text{and} \quad V_{n+1}^{(x,y)} = |y - \mu_0(x)|.$$

The conformal prediction interval (4), defined at a point $x \in \mathbb{R}^d$, reduces to

$$\widehat{C}_n(x) = \mu_0(x) \pm \text{Quantile}\Big(1 - \alpha; \big\{|Y_i - \mu_0(X_i)|\big\}_{i=1}^n \cup \{\infty\}\Big), \tag{A.2}$$

and by Theorem 1 it has coverage at least $1 - \alpha$, conditional on $(X_1^0, Y_1^0), \ldots, (X_{n_0}^0, Y_{n_0}^0)$. This holds because, when we treat $\mu_0$ as fixed (i.e., condition on $(X_1^0, Y_1^0), \ldots, (X_{n_0}^0, Y_{n_0}^0)$), the scores $V_1^{(x,y)}, \ldots, V_{n+1}^{(x,y)}$ are exchangeable for $(x, y) = (X_{n+1}, Y_{n+1})$, since $(X_1, Y_1), \ldots, (X_{n+1}, Y_{n+1})$ are.

As split conformal prediction can be seen as a special case of conformal prediction, in which the regression function $\mu_0$ is treated as fixed, Corollary 1 also applies to the split scenario, and ensures that the band defined for $x \in \mathbb{R}^d$ by

$$\widehat{C}_n(x) = \mu_0(x) \pm \text{Quantile}\bigg(1 - \alpha; \sum_{i=1}^n p_i^w(x)\delta_{|Y_i - \mu_0(X_i)|} + p_{n+1}^w(x)\delta_\infty\bigg), \tag{A.3}$$

with the probabilities as in (6), has coverage at least $1 - \alpha$, conditional on $(X_1^0, Y_1^0), \ldots, (X_{n_0}^0, Y_{n_0}^0)$.

Likewise, Theorem 2 carries over to the split conformal scenario, the weighted conformal prediction interval in (11) now simplifying to

$$\widehat{C}_n(x) = \mu_0(x) \pm \text{Quantile}\bigg(1 - \alpha; \sum_{i=1}^n p_i^w\big(Z_1, \ldots, Z_n, (x,y)\big)\delta_{|Y_i - \mu_0(X_i)|} +$$

$$p_{n+1}^w\big(Z_1, \ldots, Z_n, (x,y)\big)\delta_\infty\bigg).$$

with the probabilities as defined in (10), and Theorem 2 ensures this has coverage at least $1 - \alpha$, conditional on $(X_1^0, Y_1^0), \ldots, (X_{n_0}^0, Y_{n_0}^0)$.

## A.4 Exponentially tilting the airfoil data

Figure A.1 visualizes the effect of the exponential tilting (8) in the airfoil data set. Only the 1st and 5th dimensions of the covariate distribution are tilted; the bottom row of Figure A.1 plots the marginal densities of the 1st and 5th covariates (estimated via kernel smoothing) before and after the tilt. The top row plots the response versus the 1st and 5th covariates, simply to highlight the fact that there is heteroskedasticity, and thus we might expect the shift in the covariate distribution to have some effect on the validity of the ordinary conformal prediction intervals.

Figure A.1: The top row plots the response in (a randomly chosen half of) the airfoil data set, versus the 1st and 5th covariates. The bottom row plots kernel density estimates for the 1st and 5th covariates, in black. Also displayed are kernel density estimates for the 1st and 5th covariates after exponential tilting (8), in blue.

## A.5 More airfoil data simulation results

**Lengths of weighted conformal intervals.** Figure A.2 conveys the same setup as Figure 1, but displays histograms of the median lengths of prediction intervals rather than empirical coverages (meaning, in each of the 5000 trials, we ran unweighted or weighted split conformal prediction to cover test points, and report the median length of the prediction intervals over the test sets). We see no differences in the lengths of ordinary split conformal intervals (top row) when there is or is not covariate shift, as expected since these two settings differ only in the distributions of their test sets, but use the same procedure and have the same distribution of the training data. We see that the oracle-weighted split conformal intervals are longer than the ordinary split conformal intervals that use an equivalent effective sample size (middle row). This is also as expected, since in the former situation, the regression function $\mu_0$ was fit on training data $D_{\text{train}}$ of a different distribution than $D_{\text{shift}}$, and $\mu_0$ itself should ideally be adjusted to account for covariate shift (plenty of methods for this are available from the covariate shift literature, but we left it unadjusted for simplicity). Lastly, we see that the random forests-weighted split conformal intervals are more variable, and in some cases, much longer, than the logistic regression-weighted split conformal intervals (bottom row, difficult to confirm visually because the bars in the histogram lie so close to the x-axis).

**Weighted conformal when there is actually no covariate shift.** Lastly, Figure A.3 compares the empirical coverages and median lengths of split conformal intervals to cover points in $D_{\text{test}}$ (no covariate shift), using the ordinary unweighted approach (in red), the logistic regression-weighted

Figure A.2: Median lengths of conformal prediction intervals, computed using 5000 different random splits of the airfoil data set. The averages of median lengths in each histogram are marked on the x-axis.

Figure A.3: Empirical coverages and median lengths from conformal prediction, on the airfoil data set, with no covariate shift.

approach (in gray), and the random forests-weighted approach (in green). The unweighted and logistic regression approaches are very similar. The random forests approach yields slightly more dispersed coverages and lengths. This is because random forests are very flexible, and in the present case of no covariate shift, the estimated weights from random forests in each repetition are in general further from constant (compared to those from logistic regression). Still, random forests must not be overfitting dramatically here, since the coverages and lengths are still reasonable.

## A.6   Alternate proof of Lemma 1

We now give an alternate proof of Lemma 1, which sheds light onto why the weighted generalization in Lemma 3 holds. The general strategy we pursue here is to condition on the unlabeled multiset of values obtained by our random variables $V_1, \ldots, V_{n+1}$, and then inspect the probabilities that the last random variable $V_{n+1}$ attains each one of these values. For simplicity, we assume that there are almost surely no ties among the scores $V_1, \ldots, V_{n+1}$, so that we can work with sets rather than multisets (our arguments apply to the general case as well, but the notation is more cumbersome).

Denote by $E_v$ the event that $\{V_1, \ldots, V_{n+1}\} = \{v_1, \ldots, v_{n+1}\}$, and consider

$$\mathbb{P}\{V_{n+1} = v_i \,|\, E_v\}, \; i = 1, \ldots, n+1.$$

Denote by $f$ the probability density function[1] of the joint sample $V_1, \ldots, V_{n+1}$. By exchangeability,

$$f(v_1, \ldots, v_{n+1}) = f(v_{\sigma(1)}, \ldots, v_{\sigma(n+1)})$$

for any permutation $\sigma$ of the numbers $1, \ldots, n+1$. Thus, for each $i$, we have

$$\mathbb{P}\{V_{n+1} = v_i \mid E_v\} = \frac{\sum_{\sigma : \sigma(n+1) = i} f(v_{\sigma(1)}, \ldots, v_{\sigma(n+1)})}{\sum_\sigma f(v_{\sigma(1)}, \ldots, v_{\sigma(n+1)})}$$

$$= \frac{\sum_{\sigma : \sigma(n+1) = i} f(v_1, \ldots, v_{n+1})}{\sum_\sigma f(v_1, \ldots, v_{n+1})}$$

$$= \frac{n!}{(n+1)!} = \frac{1}{n+1}.$$

This shows that the distribution of $V_{n+1} | E_v$ is uniform on the set $\{v_1, \ldots, v_{n+1}\}$, i.e.,

$$V_{n+1} | E_v \sim \frac{1}{n+1} \sum_{i=1}^{n+1} \delta_{v_i},$$

and it follows immediately that

$$\mathbb{P}\left\{ V_{n+1} \leq \mathrm{Quantile}\left( \beta; \frac{1}{n+1} \sum_{i=1}^{n+1} \delta_{v_i} \right) \,\Big|\, E_v \right\} \geq \beta.$$

On the event $E_v$, we have $\{V_1, \ldots, V_{n+1}\} = \{v_1, \ldots, v_{n+1}\}$, so

$$\mathbb{P}\left\{ V_{n+1} \leq \mathrm{Quantile}\left( \beta; \frac{1}{n+1} \sum_{i=1}^{n+1} \delta_{V_i} \right) \,\Big|\, E_v \right\} \geq \beta.$$

Because this true for any $v$, we can marginalize to obtain

$$\mathbb{P}\left\{ V_{n+1} \leq \mathrm{Quantile}\left( \beta; \frac{1}{n+1} \sum_{i=1}^{n+1} \delta_{V_i} \right) \right\} \geq \beta,$$

which, as argued in (A.1), is equivalent to the desired lower bound in the lemma. (The upper bound follows similarly.)

## A.7   Proof of Lemma 3

We follow the same general strategy in the alternate proof of Lemma 1 in Section A.6. As before, we assume for simplicity that $V_1, \ldots, V_{n+1}$ are distinct almost surely (but the result holds in the general case as well).

Denote by $E_z$ the event that $\{Z_1, \ldots, Z_{n+1}\} = \{z_1, \ldots, z_{n+1}\}$, and denote $v_i = \mathcal{S}(z_i, z_{1:(n+1)})$, for $i = 1, \ldots, n+1$. Let $f$ denote the density function of the joint sample $Z_1, \ldots, Z_{n+1}$. For each $i$,

$$\mathbb{P}\{V_{n+1} = v_i \mid E_z\} = \mathbb{P}\{Z_{n+1} = z_i \mid E_z\} = \frac{\sum_{\sigma : \sigma(n+1) = i} f(z_{\sigma(1)}, \ldots, z_{\sigma(n+1)})}{\sum_\sigma f(z_{\sigma(1)}, \ldots, z_{\sigma(n+1)})},$$

and as $Z_1, \ldots, Z_{n+1}$ are weighted exchangeable,

$$\frac{\sum_{\sigma : \sigma(n+1) = i} f(z_{\sigma(1)}, \ldots, z_{\sigma(n+1)})}{\sum_\sigma f(z_{\sigma(1)}, \ldots, z_{\sigma(n+1)})} = \frac{\sum_{\sigma : \sigma(n+1) = i} \prod_{j=1}^{n+1} w_j(z_{\sigma(j)}) \cdot g(z_{\sigma(1)}, \ldots, z_{\sigma(n+1)})}{\sum_\sigma \prod_{j=1}^{n+1} w_j(z_{\sigma(j)}) \cdot g(z_{\sigma(1)}, \ldots, z_{\sigma(n+1)})}$$

$$= \frac{\sum_{\sigma : \sigma(n+1) = i} \prod_{j=1}^{n+1} w_j(z_{\sigma(j)}) \cdot g(z_1, \ldots, z_{n+1})}{\sum_\sigma \prod_{j=1}^{n+1} w_j(z_{\sigma(j)}) \cdot g(z_1, \ldots, z_{n+1})}$$

$$= p_i^w(z_1, \ldots, z_{n+1}).$$

In other words,

$$V_{n+1} | E_z \sim \sum_{i=1}^{n+1} p_i^w(z_1, \ldots, z_{n+1}) \delta_{v_i},$$

which implies that

$$\mathbb{P}\left\{ V_{n+1} \leq \text{Quantile}\left( \beta; \sum_{i=1}^{n+1} p_i^w(z_1, \ldots, z_{n+1})\delta_{v_i} \right) \,\middle|\, E_z \right\} \geq \beta.$$

This is equivalent to

$$\mathbb{P}\left\{ V_{n+1} \leq \text{Quantile}\left( \beta; \sum_{i=1}^{n+1} p_i^w(Z_1, \ldots, Z_{n+1})\delta_{V_i} \right) \,\middle|\, E_z \right\} \geq \beta,$$

and after marginalizing,

$$\mathbb{P}\left\{ V_{n+1} \leq \text{Quantile}\left( \beta; \sum_{i=1}^{n+1} p_i^w(Z_1, \ldots, Z_{n+1})\delta_{V_i} \right) \right\} \geq \beta.$$

Finally, as in (A.1), this is equivalent to the claim in the lemma.

## A.8    Proof of Theorem 2

Abbreviate $V_i = V_i^{(X_{n+1}, Y_{n+1})}$, $i = 1, \ldots, n+1$. Note $Y_{n+1} \in \widehat{C}_n(X_{n+1})$ if and only if

$$V_{n+1} \leq \text{Quantile}\left( 1 - \alpha; \sum_{i=1}^{n} p_i^w(Z_1, \ldots, Z_{n+1})\delta_{V_i} + p_{n+1}^w(Z_1, \ldots, Z_{n+1})\delta_\infty \right),$$

and applying Lemma 3 gives the result.

## A.9    Proof of Corollary 1

We return to the case of covariate shift, and show that Corollary 1 follows from the general weighted conformal result. By Lemma 2, we know that the independent draws $Z_i = (X_i, Y_i)$, $i = 1, \ldots, n+1$ are weighted exchangeable with $w_i \equiv 1$ for $i = 1, \ldots, n$, and $w_{n+1}((x, y)) = w(x)$. In this special case, the probabilities in (10) simplify to

$$p_i^w(z_1, \ldots, z_{n+1}) = \frac{\sum_{\sigma : \sigma(n+1)=i} w(x_i)}{\sum_\sigma w(x_{\sigma(n+1)})} = \frac{w(x_i)}{\sum_{j=1}^{n+1} w(x_j)}, \; i = 1, \ldots, n+1,$$

in other words, $p_i^w(Z_1, \ldots, Z_n, (x, y)) = p_i^w(x)$, $i = 1, \ldots, n+1$, where the latter are as defined in (6). Applying Theorem 2 gives the result.

## Footnotes

[1] More generally, $f$ may be the Radon-Nikodym derivative with respect to an arbitrary base measure.