[Reviews · NeurIPS 2019]

Reviewer 1



The presented paper generalizes the standard setting in regression via conformal prediction in order to adjust for covariate shift between train and test sets (i.e. while the conditional probability P_{Y|X} is constant, the covariate marginal P_X changes from train to test set). Although the structure of the paper seems somewhat uncommon, it is very easy to follow and gives excellent explanations of assumptions, interpretations and theoretical and empirical results. Proofs are supplied in adequate detail. A broader look on the context and importance of the generalized exchangeability is provided, indicating the paper's significance beyond the special case of covariate shift. I would consider the contributions of this paper as very significant. Conformal Prediction is especially relevant in safety sensitive situations (such as the medical domain, or autonomous driving), but in these settings covariate shift is extremely common. To properly deploy ML algorithms in these settings, covariate shift has to be taken care of - this paper provides both the algorithmic and the theoretical basis for this. Some (minor) critic points: - a theoretical analysis of the approximated likelihood ratio would have been very helpful; in practice, the true likelihood ratio will not be known, and although the authors give a practical way of dealing with this situation, the theoretical properties were not established - it would have been interesting to see, whether and/or how the proposed method translates to the classification setting, instead of only regression - the empirical evaluation is somewhat limited. A more rigorous evaluation on a more diverse range of data sets would have been helpful. - code could have easily been provided via anonymized links or in the supplement

Reviewer 2



Summary Conformal prediction is a distribution-free method for confidence region construction, which resembles the jackknife procedure. This paper addresses one failure of the exchangeability assumption, which is at the core of the theory underlying conformal prediction. The essence of the proposed adjustment to the theory and the modification of confidence interval construction is the addition of importance weights, which help deal with differences in distribution of the observed sample, e.g. due to covariate shift in supervised learning tasks. Authors numerically compare the classical procedure to the weighted prediction. The experiments also compare weight oracle against weights estimated via a contrast-based density estimation approach.

Reviewer 3



[Replies to author feedback] I thank the authors for the provided answers, in particular the extent to which the extensions concerning sample-wise covariate may prove useful. Originality: the problem of considering test distributions different from the input distributions one is not new, and the originality of the paper mainly lies in showing that it can be achieved for conformal prediction, provided we can find a map from the training to the test distribution (in this case, by estimating a likelihood ratio). I also missed the discussion of relations with techniques such as transfer learning, and also importance sampling ideas (admittedly less connected, but I think relevant nevertheless) Clarity: the paper is quite clear. Significance: this is maybe the weakest point of the paper. In particular: - The experiments are more a proof-of-concept than a demonstration that the method is competitive and applicable. In particular, conformal prediction having been designed to perform instance-wise covered predictions, it is unclear how much the idea of storing shifted observation is realistic? Or could we detect the drift incrementally? - The interest of the last part, substantiated by Theorem 2, is unclear to me from a practical point of view. I understand this generalizes previous results, but how practical is a setting where each data point may be "drifted" or may come from a different distribution? How could we solve the estimation problem in such a setting? In short, what is the added value of this generalisation (beyond being a generalisation)? Typos: * P2, L3 after top: for multiple instances OF the same

[Author Response · NeurIPS 2019]

We thank the reviewers for their thorough comments. One common comment from the reviewers is that they would like to see experiments on more than one data set. If the reviewers find this point to be critical, we would be happy to add another experiment to the supplement in the camera-ready version (we have already completed additional experiments, but left them out because the results were very similar to those on the airfoil data—the theoretical guarantees we provide are exact in finite sample without any distributions assumptions, so in the absence of estimating the likelihood ratio for the weights, there should be no surprises when running our weighted conformal prediction on a new data set). Below we respond to some of the specific comments and questions in the reviews. Square brackets [...] indicate the reviewer's comments and our replies follow.

**Reviewer #2**

Thank you for your detailed and positive feedback.
[...analysis of the effects of approximating the likelihood ratio...] Great question; if we have a guarantee on estimating $w(x) = \mathrm{d}\tilde{P}_X(x)/\mathrm{d}P_X(x)$ in the $\ell_\infty$ (essential supremum) norm, i.e., if we have bound on

$$\|\hat{w} - w\|_\infty = \text{ess sup}_{x \in \mathbb{R}^d} |\hat{w}(x) - w(x)|,$$

then we can show how to translate this into an approximate coverage guarantee on the conformal prediction intervals. We can include a corollary to this effect in the main paper, with a proof in the supplement. This would of course be a first step: an $\ell_\infty$ accuracy guarantee may be stringent, but a more careful analysis should possible in more specialized situations, out of the scope of the current paper.
[...classification...] This is a great point. The same methodology extends seamlessly to classification (only the notation changes) and we can point this out in the camera-ready version.
[...code...] We apologize for not including an anonymized link. Our code is publicly available on github and we will add the link as soon as the paper can be de-anonymized.

**Reviewer #4**

Thank you for your detailed and helpful review.
[...additional experiments...] Please see our comment at the top of the page.
[...analysis of the effects of approximating the likelihood ratio...] Please see our response to Reviewer #2.
[...computational tractability...] Yes, in general the complexity may be combinatorial. However, as mentioned in the paper, fortunately for covariate shift (and a few other scenarios mentioned in the discussion), the calculation is easier.

**Reviewer #5**

Thank you for your detailed and astute review.
[...transfer learning...] As far as we can tell, this is not directly related the goal of our paper, but as transfer learning is often discussed alongside covariate shift, it would interesting to consider this connection; space permitting we can include a comment in the camera-ready version.
[...importance sampling...] These ideas are intimately connected to our work, and form the basis for much of the methodology in covariate shift, though the modern covariate shift literature simply tends to use different language than the older importance sampling literature in statistics. One could use importance sampling to approximately fix the incompatibility of test and training distributions in conformal prediction (sample from a batch of test points with probabilities that are proportional to their likelihood in the training sample). Our methodology is in essence a *direct* way of doing this, without requiring subsampling, and is preferable since subsampling would introduce additional variance. We can include this interpretation in the camera-ready if accepted.
[...experiments...] Beyond what we mentioned at the top of this page, the experiment is indeed proof-of-concept. But we do believe it shows the method to be practically applicable; we have investigated several angles of the problem, including what happens when we still estimate a likelihood ratio and there is actually no covariate shift (results given in the supplement). Regarding testing whether our method is "competitive", as far as we know, there are no other methods that produce provably valid prediction intervals (with distribution-free, finite-sample coverage guarantees) under covariate shift, so we do not know of competing methods that we might compare against.
[...detect drift incrementally...] This is an interesting question. There is orthogonal work on using conformal prediction to detect a change in distribution: see "Inductive Conformal Martingales for Change-Point Detection". Future work could consider sequential versions of a problem of both detecting covariate shift and producing appropriately corrected prediction intervals, combining the insights of our paper with theirs.
[...generalization to each point coming from different distribution...] While we do do not know of any examples or applications that need this full generality, it is very plausible that the test points lie in (multiple) unknown clusters, with each cluster coming from a different distribution shift. This type of distribution could possibly be detected by mixture methods, correcting for each cluster individually. We are still exploring the full power of the general technique and will continue to do so in future work, but it is outside the scope of this short paper.

[Meta-Review · NeurIPS 2019]

The reviewers agree that the contributions of the paper are novel, significant and clearly exposed. The experimental part could be more developed.